# The Use of the Deep Brachial Artery as the Recipient Artery for Free Perforator Flap Transfer: An Anatomic Study and Clinical Applications

**DOI:** 10.3390/medicina59061087

**Published:** 2023-06-05

**Authors:** Hidehiko Yoshimatsu, Ryo Karakawa, Yuma Fuse, Tomoyuki Yano, Satoru Muro, Keiichi Akita

**Affiliations:** 1Cancer Institute Hospital of the Japanese Foundation for Cancer Research, Department of Plastic and Reconstructive Surgery, 3-8-31 Ariake, Koto-ku, Tokyo 135-8550, Japan; 2Department of Clinical Anatomy, Tokyo Medical and Dental University (TMDU), 1-5-45 Yushima, Bunkyo-ku, Tokyo 113-8510, Japan

**Keywords:** deep brachial artery, cadaver study, superficial circumflex iliac artery perforator flap, perforator flap

## Abstract

*Background and Objectives:* Soft tissue reconstruction after sarcoma ablation in the posterior aspect of the upper arm has been commonly addressed using the pedicled latissimus dorsi musculo-cutaneous flap. The use of a free flap for coverage of this region has not been reported in detail. The goal of this study was to characterize the anatomical configuration of the deep brachial artery in the posterior upper arm and assess its clinical utility as a recipient artery for free-flap transfers. *Materials and Methods:* In total, 18 upper arms from 9 cadavers were used for anatomical study to identify the deep brachial artery’s origin and point of crossing the *x*-axis, which was set from the acromion to the medial epicondyle of the humerus. Measurements of the diameter were taken at each point. The anatomic findings of the deep brachial artery were employed clinically in the reconstruction of the posterior upper arm after sarcoma resection using free flaps in 6 patients. *Results:* The deep brachial artery was found in all specimens between the long head and the lateral head of the triceps brachii muscle, and it crossed the *x*-axis at an average distance of 13.2 ± 2.9 cm from the acromion, with an average diameter of 1.9 ± 0.49 mm. In all 6 clinical cases, the superficial circumflex iliac perforator flap was transferred to cover the defect. The average size of the recipient artery, the deep brachial artery, was 1.8 mm (range, from 1.2 to 2.0 mm). The average diameter of the pedicle artery, the superficial circumflex iliac artery, was 1.5 mm (range, from 1.2 to 1.8 mm). All flaps survived completely with no postoperative complications. *Conclusions:* The deep brachial artery can be a reliable recipient artery in free-flap transfers for posterior upper arm reconstruction, given its anatomical consistency and sufficient diameter.

## 1. Introduction

Sarcomas are a rare type of cancer, accounting for only about 1% of all adult cancers and 15% of pediatric cancers [1]. They can arise in any part of the body that contains connective tissue, including the bones, cartilage, fat, muscles, blood vessels, and nerves. There are several subtypes of sarcoma, and the treatment and prognosis depend on the specific type, location, and stage of the cancer. The treatment of sarcomas often involves surgical removal of the affected tissue, which can result in significant defects that require reconstruction [2]. However, reconstruction after sarcoma treatment presents a unique challenge due to the frequent need for postoperative radiotherapy and/or chemotherapy. Complications such as partial flap necrosis can delay these treatments, potentially negatively impacting the patient’s overall survival. Therefore, a flap with a low risk of complications is ideal for use in sarcoma treatment. 

Pedicled island flaps are commonly utilized for addressing soft tissue defects in the trunk and extremities [3,4,5,6]. These flaps, which include pedicled propeller flaps and pedicled transposition flaps, are advantageous because they do not necessitate vessel anastomosis, making them less resource-intensive and quicker to perform than free-flap transfer. Nonetheless, due to their restricted mobility, there can be difficulties in placing pedicled flaps, particularly when rotating or exerting pressure on the pedicle.

Soft tissue reconstruction after sarcoma ablation in the posterior aspect of the upper arm has been commonly addressed using the pedicled latissimus dorsi (LD) musculocutaneous flap [7,8]. However, the wide range of motion of the shoulder joint and the limited length of the pedicle can cause marginal loss of the distal flap. The use of a free flap for coverage of this region has not been reported in detail. This was due to the fact that it is difficult to find a sizable recipient artery in this region. The use of perforator flaps has recently become common due to their lower donor site morbidity [9,10,11,12]. Now that the harvesting techniques of perforator flaps have been well described, the success rate of the free-flap transfer predominantly depends on the recipient vessel, especially the artery. The goal of this study was to characterize the anatomical configuration of the deep brachial artery in the posterior upper arm and assess its clinical utility as a recipient artery for free-flap transfers.

## 2. Materials and Methods

### 2.1. Anatomic Study

In total, 18 upper arms from 9 cadavers (1 male and 8 female) with an average age of 84.3 years (range, 70 to 94) were donated to our department. The average body weight of the cadavers was 38.4 kg (range, 26 to 52 kg). The donation document format was congruent with the Japanese law entitled “The Act on Body Donation for Medical and Dental Education” (Act No. 56 of 1983). Before their death, all donors voluntarily declared that their remains would be donated as materials for education and study. At that time, the purpose and methods of using body donor corpses were explained, and informed consent was obtained. After their death, we explained the informed consent to the bereaved families and confirmed that there was no opposition. All cadavers were fixed via arterial perfusion with 8% formalin and preserved in 30% alcohol. The study was approved by the Board of Ethics at the Tokyo Medical and Dental University (approval number: M2018-006). All methods were performed following the relevant guidelines and regulations. The study demographics are shown in Table 1.

The deep brachial artery is the first branch of the brachial artery, arising from its posterior aspect, and it travels a path that lies between the long and medial heads of the triceps brachii. In this series, the specimens were placed in the prone position and the acromion and the medial epicondyle of the humerus were marked. An *x*-axis was designed from the acromion to the medial epicondyle of the humerus (Figure 1). An incision was made to the muscle layer, where the long and medial heads of the triceps brachii were identified. A muscle retractor was placed between the two heads of the triceps brachii, and the brachial artery and its first branch, the deep brachial artery, were carefully identified using surgical loupes with 2.5× magnification. The location of the origin of the deep brachial artery and the point where it crossed the *x*-axis were recorded (Figure 2). The diameters of the artery were measured at each point.

### 2.2. Clinical Applications

The anatomic findings of the deep brachial artery were employed clinically in the reconstruction of the posterior upper arm after sarcoma resection using free flaps. Institutional Review Board (IRB) approval was obtained at the Cancer Institute Hospital of the Japanese Foundation for Cancer Research (#2021-GB-044). This retrospective analysis included a systematic review of all operation reports and clinical follow-up data of 6 consecutive sarcoma patients who underwent immediate reconstruction using the deep brachial artery as the recipient artery for free-flap transfer between June 2021 and September 2022. There were 4 male patients and 2 female patients. The average age of the patients was 56.5 years (range, 27 to 79 years), and the average body mass index of the patients was 21.7 (range, 17.8 to 23.4). Ablation of the tumor was performed by surgical oncologists with a wide safety margin. All defects after sarcoma resection were in the posterior aspect of the upper arm. The resection and reconstruction were performed with the patient in either a prone or lateral position. The details of patients are presented in Table 2. 

We implemented the three-step ICG angiography protocol to ensure a secure and reliable transfer of free flaps [13]. The protocol involved the following steps:(1)Intravenous administration of 1.0 mL of ICG (Diagnogreen 0.25%; Daiichi Pharmatical, Tokyo, Japan) to verify flap perfusion after flap elevation but prior to pedicle transection. A near-infrared fluorescence imaging device (LIGHTVISION; Shimadzu Corporation, Kyoto, Japan) was used to evaluate flap perfusion. Any poorly highlighted areas observed in the flap after 3 min of ICG administration were deemed ischemic or congestive. If no bleeding was observed during a prick test using a 23-gauge needle, the flap was considered ischemic, and the continuity and patency of the pedicle artery were assessed. If continuous bleeding was seen during the prick test, congestion of the flap was suspected.(2)Intravenous administration of 1.0 mL of ICG after completion of vessel anastomoses to confirm patency of anastomosis sites.(3)Intravenous administration of 1.0 mL of ICG after inset of the flap. Any poorly highlighted areas observed 3 min after ICG administration were deemed ischemic or congestive. If no bleeding was observed during a prick test using a 23-gauge needle, the flap was considered ischemic, and the patency of the arterial anastomosis site was assessed and addressed if necessary. If continuous bleeding was observed during the prick test, congestion of the flap was suspected. If excessive tension to the flap or the pedicle or kinking of the pedicle was found, it was mitigated. ICG angiography was performed again after a minimal interval of 20 min.

Postoperatively, a vasodilator was infused continuously for 5 days (10 μg/day lipo-prostaglandin E1 (Prostandin; Ono, Osaka, Japan)).

## 3. Results

### 3.1. Anatomic Findings

The deep brachial artery was identified in all specimens between the long head and the lateral head of the triceps brachii muscle as the first branch of the brachial artery. In all specimens, it continued along the radial sulcus, which is a groove for the radial nerve. Along this path, the artery was covered by the lateral head of the triceps brachii until it reached the lateral side of the arm. At that point, it passed through the lateral intermuscular septum and descended further between the brachioradialis and the brachialis, eventually arriving at the front of the lateral epicondyle of the humerus. Finally, the artery terminated by joining with the radial recurrent artery via anastomosis. In all specimens, the first portion of the deep brachial artery ran along the radial nerve.

The average distance from the acromion to its takeoff from the brachial artery along the *x*-axis was 9.8 ± 2.5 cm (range, 6.5 to 16.0 cm), and the average diameter was 2.7 ± 0.71 mm (range, 2.0 to 4.0 mm) at this point. The deep brachial artery crossed the *x*-axis at an average distance of 13.2 ± 2.9 cm (range, 9.0 to 21.0 cm) from the acromion. The average diameter of the deep brachial artery at this distal point was 1.9 ± 0.49 mm (range, 1.0 to 3.0 mm). These findings are shown in Table 1 and depicted in Figure 2.

### 3.2. Clinical Results

In all six cases, a free superficial circumflex iliac perforator (SCIP) flap was used to cover the defect. The average length and width of the SCIP flap were 17.2 cm (range, from 15.0 to 20.0 cm) and 9.8 cm (range, from 2.0 to 4.0 cm), respectively. The average length of the pedicle was 2.8 cm (range, from 2.0 to 4.0 cm). The flaps were harvested at the same time as the tumor resection to shorten the operative time. The average size of the recipient artery, the deep brachial artery, was 1.8 mm (range, from 1.2 to 2.0 mm). The average diameter of the pedicle artery, the superficial circumflex iliac artery (SCIA), was 1.5 mm (range, from 1.2 to 1.8 mm), and all arterial anastomoses were performed in an end-to-end fashion. In all cases, the flap was sufficiently highlighted with ICG angiography performed after flap elevation, after anastomosis, and finally after the flap inset. The average follow-up period was 8.7 months (range, from 4 months to 18 months). All flaps survived completely with no postoperative complications. There was no postoperative numbness or weakening of the muscles innervated by the radial nerve. These findings are summarized in Table 2.

### 3.3. Case

A 48-year-old woman was diagnosed with sarcoma of the posterior upper arm (Figure 3, left). After a wide resection, a 16 × 10 cm defect was noted (Figure 3, right). A 17 × 10 cm SCIP flap was elevated (Figure 4). The deep brachial artery was found and dissected between the long head and the lateral head of the triceps brachii muscle. The deep brachial artery was cut at a distal point and the vessel was pulled up between the muscles for easier anastomosis (Figure 5). The diameter of the deep brachial artery was 1.2 mm, and the diameter of the pedicle artery (SCIA) of the flap was 1.5 mm. The arterial anastomosis was performed in an end-to-end fashion using 9-0 nylon sutures. The diameters of the superficial circumflex iliac vein and the vena comitans of the SCIA were both 1.5 mm. Both flap veins were anastomosed to the venae comitantes of the deep brachial artery in an end-to-end fashion using a 1.5 mm coupling device. The diameters of the venae comitantes of the deep brachial artery were both 1.5 mm. The defect was covered with the SCIP flap, and the flap survived completely with a satisfactory contour (Figure 6). The donor site healed uneventfully (Figure 7).

## 4. Discussion

After the introduction of free flaps with less morbidity such as the scapular flap and the parascapular flap, more than 20 years have passed since the first report of the perforator flap; the refinements of perforator flap harvesting have been well described [14,15,16,17,18]. The donor site morbidity is minimized when the flap is harvested with its pedicle artery at the perforator level, which often results in a small pedicle artery of around 1.0 mm. This small pedicle becomes an advantage once the technical challenge of anastomosis per se is overcome because the pedicle can be anastomosed to recipient vessels of any size. It can be anastomosed in an end-to-end fashion to any small recipient perforator artery. If no small recipient vessels can be found, an end-to-side anastomosis can be performed to larger source vessels. In addition, the perfusion of a flap can be confirmed using indocyanine green angiography [13]. With all these advancements, it is not an overstatement to argue that, in the present era, the success of a perforator flap transfer depends on the recipient vessels. 

As with flap pedicle arteries, the use of perforators as recipient vessels was introduced and gained popularity [19]. Recent modalities such as ultrasonography allow more precise identification of these perforators, but it is always reassuring to have anatomical knowledge regarding where to look for suitable recipient arteries in all regions of the body. This led us to look into the use of the deep brachial artery as the recipient artery in free-flap reconstruction of the posterior upper arm. 

In our cadaver study, the average diameter of the deep brachial artery was 2.7 mm at its origin and 1.9 mm at its distal region. The deep brachial artery is a dispensable artery, and these results indicate that the recipient artery diameter can be adjusted depending on where you cut the deep brachial artery. In our clinical cases in which we chose the SCIP flap for coverage, the deep brachial artery was found between the long head and the lateral head of the triceps brachii muscle in all cases and was cut distally, providing us with an average diameter of 1.8 mm. This was an optimal size match for the SCIA, which has an average diameter of 1.5 mm. The use of the deep brachial artery allowed anastomosis within the defect, and thus a flap with a long pedicle was not necessary. Having a good understanding of the deep brachial artery’s anatomy significantly reduces the burden of finding suitable recipient vessels, particularly in the posterior upper arm where recipient arteries are scarce. One pitfall that should be prevented when using the deep brachial artery as the recipient artery is damaging the radial nerve. As it runs deep between the long head and the lateral head of the triceps brachii muscle, the deep brachial artery runs along the radial nerve; special caution should be taken when dissecting out the artery.

While local flaps are a relatively simple solution, they do have limitations when it comes to repairing extensive and deep tissue damage, such as in the case of reconstruction after sarcoma resection [20]. The tension caused by such procedures can lead to postoperative wound dehiscence and secondary contracture, which in turn may result in undesirable aesthetic and functional outcomes [21].

Since the introduction of the perforator flap concept, the use of pedicled island flaps has also become more widespread. Compared to traditional local flaps, these flaps offer greater mobility due to their ability to rotate and transpose [3,4,5,6]. Using a pedicled island flap for defect coverage can be accomplished in a shorter operative time without the need for an operative microscope, and the postoperative care is typically less intensive than that of free-flap transfer. However, because the flap is anchored to the pedicle origin, there is a higher risk of pedicle compression, particularly in cases involving flap rotation, which usually occurs during the final stage of reconstruction after the flap is inserted into the defect.

In a prior retrospective study, the reconstruction of the upper trunk and shoulder girdle after oncologic resection was investigated to determine the correlation between the need for pedicled or free-flap reconstruction and tumor and defect characteristics [22]. However, this was a descriptive study that did not compare the reconstruction methods while aligning tumor and patient characteristics. Therefore, it remains unclear whether pedicled or free-flap reconstruction yields better short-term outcomes.

Following its introduction in 2004, the SCIP flap has gained acceptance in Asia, primarily for reconstructing lower extremities in diabetes patients, for trauma reconstruction, and for addressing small defects resulting from the resection of malignant skin tumors such as melanoma [9,10,23,24]. Although there have been limited case series involving the use of SCIP flaps for sarcoma reconstruction, the attributes of the flap suggest it to be an ideal option for reconstruction after sarcoma resection [25,26].

The SCIP flap harvest leaves a discreet scar in a concealed region with minimal functional morbidity. Unlike other perforator flaps, such as the anterolateral thigh flap, the thoracodorsal artery perforator flap, the profunda femoris artery perforator flap, and the medial sural artery perforator flap, the donor site of the SCIP flap can be directly closed even after harvesting a flap with a width greater than 12 cm. In most cases, skin grafting of the donor site is unnecessary, precluding delayed healing of the donor site, which could postpone postoperative sarcoma treatment. Furthermore, the skin paddle has profuse laxity perpendicular to its long axis, allowing the width to be stretched during flap inset without any congestion concerns.

The SCIP pedicle can be anastomosed to recipient vessels of any size, resulting in a high percentage of anastomoses performed within the defect, as seen in 97% of our sarcoma case series [26]. This unique characteristic of the SCIP flap allows fresh and usable recipient vessels to be found almost always within the defect after sarcoma resection. With a diameter of approximately 1.0 mm, the SCIA permits the use of nearly any small perforator and its vena comitans, if found. If small recipient vessels are unavailable, surgeons can perform an end-to-side anastomosis to larger source vessels, which significantly reduces the stress associated with attempting to locate adequate recipient vessels.

The greatest disadvantage of the SCIP flap is its short pedicle. In our clinical case series, the average length of the pedicle was 2.8 cm (range, from 2.0 to 4.0 cm). However, the use of the deep brachial artery as the recipient artery compensated for this shortcoming; the deep brachial artery could be dissected out distally from its takeoff from the brachial artery and cut at a distal point to make up for the short flap pedicle. By this long dissection of the deep brachial artery and pulling it up between the long and medial heads of the triceps brachii, anastomosis could be performed in a superficial layer.

The primary limitation of this study was the small number of clinical cases. In order to provide a more accurate assessment of the effectiveness of the deep brachial artery as a recipient artery, additional research with a larger patient cohort is necessary. It is worth noting that all cases included in this study dealt with reconstruction after sarcoma removal, meaning the defects were considered “fresh”. However, if performing a reconstruction after trauma, the zone of injury would need to be taken into account, and the perforator-to-perforator method might be more challenging to implement.

There has been no report of an anatomical investigation focusing on the use of the deep brachial artery as the recipient vessel. This report clarified the presence, size, and location of the deep brachial artery, which allows safe and reliable free-flap transfer to the posterior upper arm. With the use of the deep brachial artery as a recipient vessel, free-flap transfers can potentially become a viable option for soft tissue reconstruction in the posterior aspect of the upper arm after sarcoma removal.

## 5. Conclusions

The deep brachial artery is a reliable recipient artery for free-flap transfers in the posterior upper arm for sarcoma reconstruction. This study provides detailed anatomical information on the deep brachial artery and its clinical application in six patients who underwent immediate reconstruction using free-flap transfer. The results of this study showed that the deep brachial artery is a suitable recipient artery for free-flap transfers in the posterior upper arm, with good blood supply and minimal morbidity. Regarding the type of free flap, the SCIP flap may be the best match considering the diameter of its pedicle. This information can help surgeons plan and perform free-flap transfers with greater precision and improve outcomes for patients undergoing sarcoma resection.

## Figures and Tables

**Figure 1 medicina-59-01087-f001:**
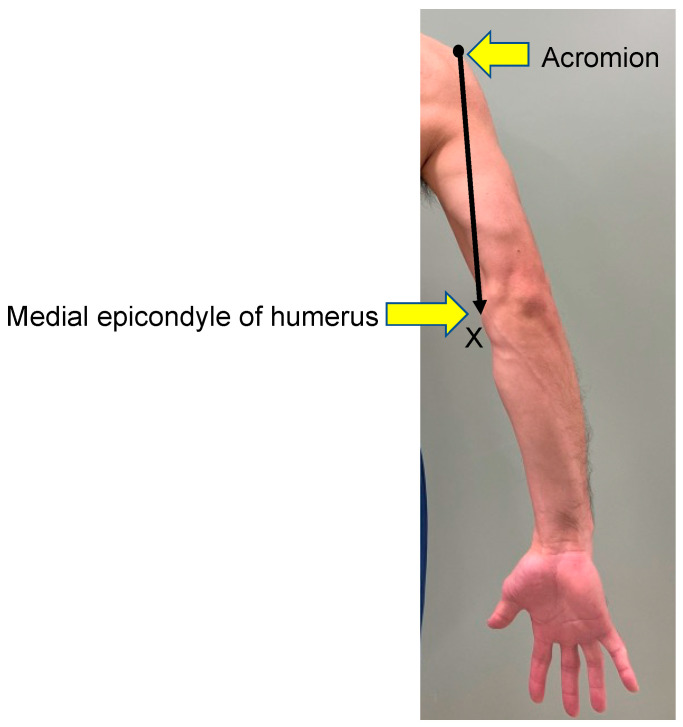
An *x*-axis was designed from the acromion to the medial epicondyle of humerus.

**Figure 2 medicina-59-01087-f002:**
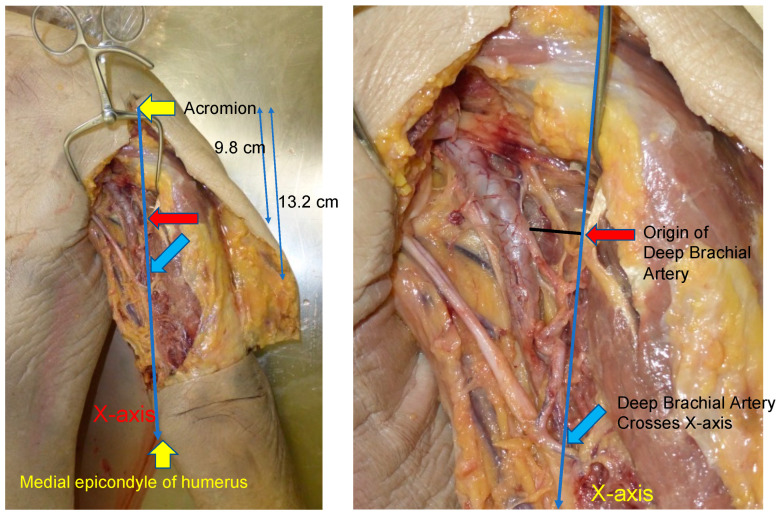
The average distance from the acromion to its takeoff (red arrow) from the brachial artery along the *x*-axis was 9.8 ± 2.5 cm. The deep brachial artery crossed the *x*-axis (blue arrow) at the average distance of 13.2 ± 2.9 cm (range, 9.0 to 21.0 cm) from the acromion.

**Figure 3 medicina-59-01087-f003:**
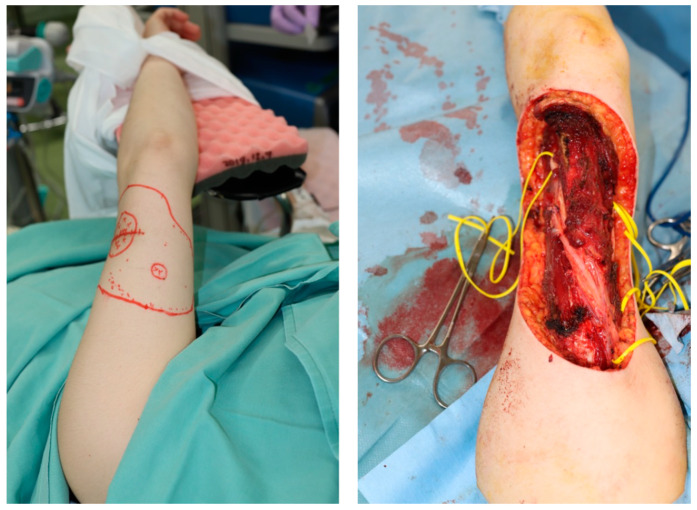
A 48-year-old woman was diagnosed with sarcoma of the posterior upper arm (**left**). Wide resection of the tumor resulted in a 16 × 10 cm defect (**right**).

**Figure 4 medicina-59-01087-f004:**
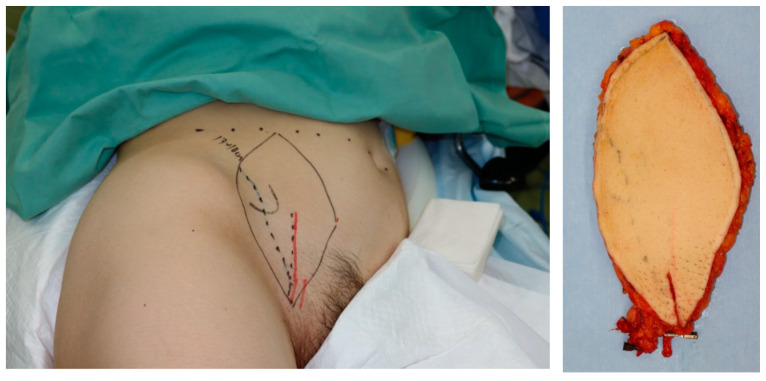
A 17 × 10 cm SCIP flap was elevated.

**Figure 5 medicina-59-01087-f005:**
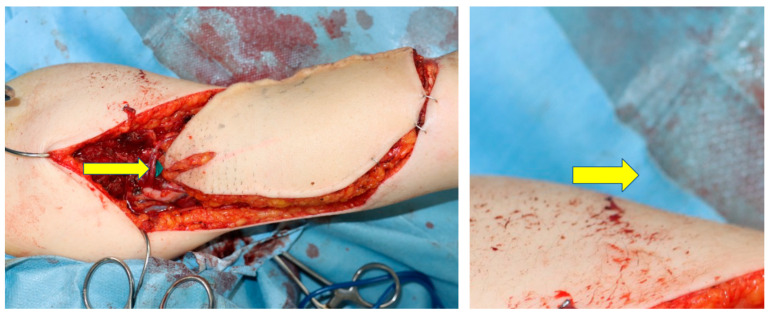
The deep brachial artery was cut at a distal point and the vessel was pulled up between the long head and the lateral head of the triceps brachii muscle for easier anastomosis (yellow arrow).

**Figure 6 medicina-59-01087-f006:**
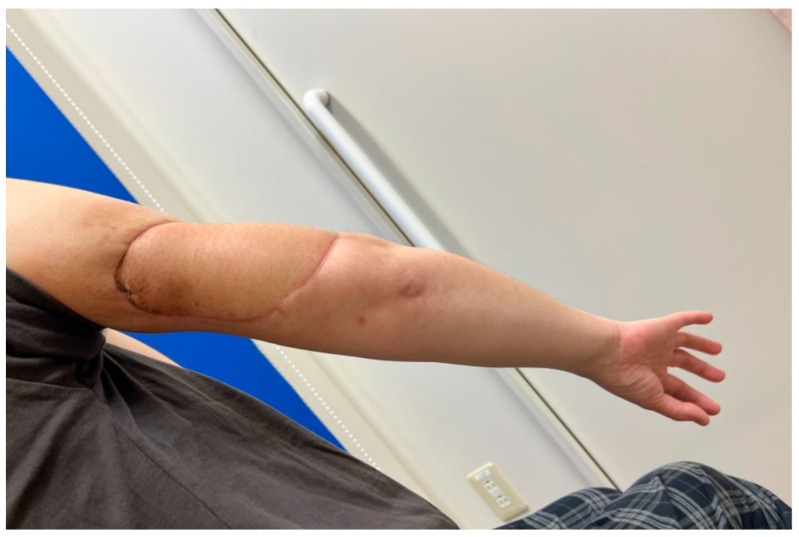
A photo taken 6 months after the surgery. The flap survived completely with a satisfactory contour.

**Figure 7 medicina-59-01087-f007:**
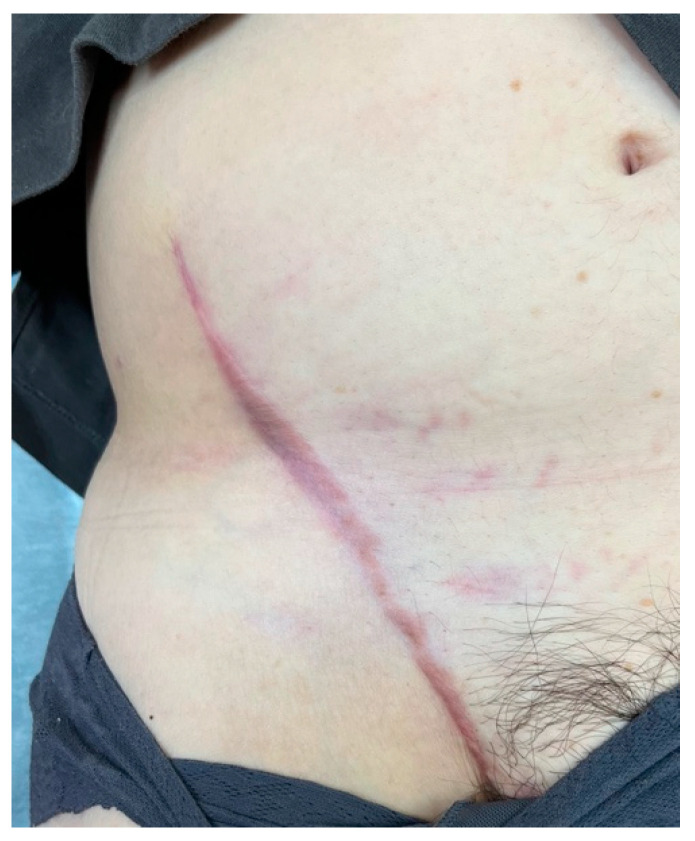
A photo of the donor site taken 6 months after the surgery.

**Table 1 medicina-59-01087-t001:** Study demographics and documented data.

					Deep Brachial Artery (at Its Origin)		Deep Brachial Artery (crossing The *x*-Axis)	
Cadaver	Age	Sex	Weight (kg)	Side	X (cm)	Diameter (mm)	X (cm)	Diameter (mm)
1	71	Female	35	Right	9.0	2.5	12.0	1.2
				Left	6.5	3.0	11.5	1.5
2	94	Female	33	Right	7.0	3.0	9.0	2.0
				Left	8.0	2.0	11.0	1.5
3	84	Female	42	Right	11.0	4.0	15.0	2.0
				Left	11.0	4.0	15.0	2.0
4	90	Female	26	Right	7.0	2.0	12.0	1.5
				Left	16.0	2.0	21.0	2.0
5	79	Female	35	Right	7.0	2.0	11.0	1.5
				Left	9.5	2.5	11.0	1.5
6	92	Female	44	Right	13.0	2.0	15.0	2.0
				Left	12.0	2.0	13.0	2.0
7	97	Female	44	Right	12.0	2.5	17.0	1.0
				Left	10.0	3.0	16.0	2.0
8	70	Female	35	Right	9.0	2.5	13.0	2.0
				Left	8.0	3.0	11.0	2.5
9	82	Male	52	Right	10.0	3.0	12.0	2.5
				Left	11.0	4.0	12.0	3.0
Mean	84.3		38.4		9.8	2.7	13.2	1.9
SD	9.8		7.7		2.5	0.71	2.9	0.49

SD, standard deviation.

**Table 2 medicina-59-01087-t002:** Patient details and reconstruction summary.

No.	Age (Years)	Sex	Body Mass Index	Defect Location	Flap Length (cm)	Flap width (cm)	Pedicle Length (cm)	Pedicle Artery	Pedicle Artery Diameter (mm)	Arterial Anastomosis Fashion	Deep Brachial Artery Diameter (mm)	Number of Veins Anastomosed	Complications	Follow-Up Length (Months)
1	46	Male	21.4	Posterior upper arm	16.0	8.0	2.0	SCIA	1.5	End to end	2.0	1	None	4
2	67	Male	23.4	Posterior upper arm	16.0	9.5	3.0	SCIA	1.2	End to end	2.5	1	None	18
3	72	Female	17.8	Posterior upper arm	15.0	9.0	3.0	SCIA	1.2	End to end	1.2	1	None	9
4	48	Female	24.3	Posterior upper arm	17.0	10.0	2.0	SCIA	1.5	End to end	1.2	2	None	9
5	79	Male	22.2	Posterior upper arm	19.0	12.0	4.0	SCIA	1.8	End to end	2.0	2	None	6
6	27	Male	21.3	Posterior upper arm	20.0	10.0	3.0	SCIA	1.5	End to end	2.0	1	None	6
Average	56.5		21.7		17.2	9.8	2.8		1.5		1.8	1.3		8.7

SCIA, superficial circumflex iliac artery.

## Data Availability

The data that support the report are available from the corresponding author, H.Y., upon reasonable request.

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
