# Peer review of "The Use of the Deep Brachial Artery as the Recipient Artery for Free Perforator Flap Transfer: An Anatomic Study and Clinical Applications"

_medicina, 2023, doi:10.3390/medicina59061087_

Round 1

Reviewer 1 Report

Good manuscript. The following comments:

Not only the LD is a good pedicled flap in the region, but also the parascapular flap. I also find it better as a free flap than the SCIP flap. It's just not as modern because it's not a perforator flap. 

I don't understand why the authors don't mention the End-to-side anastomosis as an alternative. It is actually a normal standard. 

The offered ICG concept is quite intensive. Do you really need it. Isn't the clinical assessment enough?

Prostaglandin infusion is a concept from the 90s. In my opinion it is overtherapy.

Please insert a picture of the donor site of the SCIP flap patient. 

The discussion should focus on the anatomical preparation part (arm). The discussion of the SCIP is actually not appropriate. It is about the arm.

Author Response

Thank you for your constructive comments concerning our manuscript “The Use of the Deep Brachial Artery as the Recipient Artery for Free Perforator Flap Transfer: An Anatomic Study and Clinical Applications” We have studied your comments carefully and made corrections, which we hope will meet your approval. We answered your questions or comments in detail in the following paragraphs. All changes to the manuscript are written in red text.

Author's Response to Reviewer Comments

ITEMIZED, POINT-BY-POINT RESPONSES TO THE COMMENTS OF THE REVIEWERS

Reviewer #1:

Comment 1: Not only the LD is a good pedicled flap in the region, but also the parascapular flap. I also find it better as a free flap than the SCIP flap. It's just not as modern because it's not a perforator flap.

Response 1: Thank you for this constructive comment. We added information and references on the scapular and the parascapular flap. ” After introduction of free flaps with less morbidity such as the scapular flap and the parascapular flap, more than 20 years have passed since the first report of the perforator flap.”

Nassif TM, Vidal L, Bovet JL, Baudet J. The parascapular flap: a new cutaneous microsurgical free flap. Plast Reconstr Surg. 1982;69(4):591-600. doi:10.1097/00006534- 198204000-00001

Sabino J, Franklin B, Patel K, Bonawitz S, Valerio IL. Revisiting the scapular flap: applications in extremity coverage for our U.S. combat casualties. Plast Reconstr Surg. 2013;132(4):577e-585e. doi:10.1097/PRS.0b013e31829f4a08

Comment 2: I don't understand why the authors don't mention the End-to-side anastomosis as an alternative. It is actually a normal standard.

Response 2: Thank you for this comment. End-to-side anastomosis is mentioned in Discussion: “This small pedicle becomes an advantage once the technical challenge of anastomosis per se is overcome because the pedicle can be anastomosed to recipient vessels of any size. It can be anastomosed in an end-to-end fashion to any small recipient perforator artery.”

Comment 3: The offered ICG concept is quite intensive. Do you really need it. Isn't the clinical assessment enough?

Response 3: Thank you for the comment. We routinely perform ICG angiography protocol for every flap procedure, and have reported it to be very helpful. In addition to the reference mentioned in the manuscript, we have a large-scale study coming on this topic.

Comment 4: Prostaglandin infusion is a concept from the 90s. In my opinion it is overtherapy.

Response 4: Thank you for the comment. Although it is purely anecdotal, we have seen improvements in postoperative course with the use of low-dose prostaglandin.

Comment 5: Please insert a picture of the donor site of the SCIP flap patient.

Response 5: Thank you for this suggestion. A photo of the donor site was added to the manuscript.

Comment 6: The discussion should focus on the anatomical preparation part (arm). The discussion of the SCIP is actually not appropriate. It is about the arm.

Response 6: Thank you for this comment. In fact, the information on SCIP was added due to a suggestion by the editors. We are willing to delete it, but will have to confirm with the editors.

Reviewer 2 Report

Congratulations to authors for their great study entitled "The Use of the Deep Brachial Artery as the Recipient Artery for Free Perforator Flap Transfer: An Anatomic Study and Clinical Applications"

In order to extend your introduction or discussion about the importance of reconstructive application of free flaps, I recommend to include the following references:

1. Sadeghi P, Pandey S. The Versatility of the Pedicled Medial Sural Artery Perforator Flap: From Simple to Its Chimeric Pattern and Clinical Experience with 37 Cases. Plast Reconstr Surg. 2023 Feb 1;151(2):345e. doi: 10.1097/PRS.0000000000009870. Epub 2022 Nov 15. PMID: 36696340.

2.Verdoy SB, Sadeghi P, Ojeda AL, Palacín Porté JA, Vinyals Vinyals JM, Barceló LH, Lluis EC, Compta XG, Diaz AT, Segú JOB. Evaluation of virtual surgical planning and three-dimensional configurations for reconstruction of maxillary defects using the fibula free flap. Microsurgery. 2022 Nov;42(8):749-756. doi: 10.1002/micr.30957. Epub 2022 Sep 14. PMID: 36102527.

Author Response

Reviewer #2:

Comment 1: In order to extend your introduction or discussion about the importance of reconstructive application of free flaps, I recommend to include the following references:

  1. Sadeghi P, Pandey S. The Versatility of the Pedicled Medial Sural Artery Perforator Flap: From Simple to Its Chimeric Pattern and Clinical Experience with 37 Cases. Plast Reconstr Surg. 2023 Feb 1;151(2):345e. doi: 10.1097/PRS.0000000000009870. Epub 2022 Nov 15. PMID: 36696340.

2.Verdoy SB, Sadeghi P, Ojeda AL, Palacín Porté JA, Vinyals Vinyals JM, Barceló LH, Lluis EC, Compta XG, Diaz AT, Segú JOB. Evaluation of virtual surgical planning and three-dimensional configurations for reconstruction of maxillary defects using the fibula free flap. Microsurgery. 2022 Nov;42(8):749-756. doi: 10.1002/micr.30957. Epub 2022 Sep 14. PMID: 36102527.

Response 1: Thank you for this comment. The suggested references were added to the manuscript.

Reviewer 3 Report

Very interesting and complete paper that includes an anatomical study in fresh cadaver and the clinical experience of the author. It provides interesting data and measurements for the dissection of the deep brachial artery, and includes nice photos of clinical cases. In addition, it includes limitations of the study at the end of the discussion, and it is well referenced, structured, and described. However, I would like to ask some questions to include in the manuscript:

- In the material and method section, can the authors specify if they check the sarcoma margins intraoperatively?

In the results section:

- Are the venous anastomoses done to the comitant or to another veins?

- How many flaps needed a second vein?

- Are all the vein anastomoses made with a coupler or with stitches?

Author Response

Reviewer #3:

Comment 1: In the material and method section, can the authors specify if they check the sarcoma margins intraoperatively?

Response 1: Thank you for this constructive comment. In our institution, ablation of the tumor was performed by surgical oncologists with a wide safety margin. Thus, the specimen was not sent to pathology intraoperatively. This information was added to Materials and Methods.

Comment 2: Are the venous anastomoses done to the comitant or to another veins?

Response 2: Thank you for the comment. The venous anastomoses were done to the vena comitant of the deep brachial artery.

Comment 3: How many flaps needed a second vein?

Response 3: Thank you for this question. Two out of 6 flaps required anastomosis of the second vein, which was suggested by the results of ICG angiography. This information was added to Table 2.

Comment 4: Are all the vein anastomoses made with a coupler or with stitches?

Response 4: All venous anastomoses were performed using the coupling device.

Round 2

Reviewer 2 Report

Congratulations! The study entitled “ The Use of the Deep Brachial Artery as the Recipient Artery for Free Perforator Flap Transfer: An Anatomic Study and Clinical Applications” is very interesting and the revisions are satisfying